



# Online Total Organic Carbon (TOC) monitoring for water and wastewater treatment plants processes and operations optimization

Céline Assmann[1], Amanda Scott[1], Dondra Biller[1]

[1]Analytical Instruments, a Division of GE Power, Boulder, Colorado, USA

*Correspondence to*: Céline Assmann, celine.assmann@ge.com; Amanda Scott, amanda.scott@ge.com

**Abstract.** Organic measurements, such as biological oxygen demand (BOD) and chemical oxygen demand (COD) were developed decades ago in order to measure organics in water. Today, these time-consuming measurements are still used as parameters to check the water treatment quality; however, the time required to generate a result, ranging from hours to days, does not allow COD or BOD to be useful process control parameters. Online Organic Carbon Monitoring allows for effective

process control because results are generated every few minutes. Though it does not replace BOD or COD measurements still required for compliance reporting, it drives smart, data-driven and rapid decision-making to improve process control and optimization or meet compliances. Thanks to the smart interpretation of generated data and the capability to now take real-time actions, municipal drinking water and wastewater treatment facility operators can positively impact their OPEX efficiencies and their capabilities to meet regulatory requirements. This paper describes how three municipal wastewater and

drinking water plants gained process insights, and determined optimization opportunities thanks to the implementation of on-line TOC monitoring.

## 1 Introduction

Growing populations and expanding industries are pulling on water resources while adding nutrients and pollutants to water sources. These facts coupled with heightened public demand for quality water at affordable prices has the water industry under

scrutiny. Whether complying with water regulations, optimizing treatment processes for saving time and money, or looking to better manage a plant during times of emergency (flood, fire, security threat, drought or industrial spill), knowing and understanding organics and organic removal can be extremely valuable. Total Organic Carbon (TOC) Monitoring is one of the most important parameters that drinking water and wastewater facilities can use to make decisions about treatment.

Measuring TOC can be critical to a water treatment facility's water quality in helping to optimize treatment processes. TOC is

useful in detecting the presence of many organic contaminants including petroleum products, organic acids like humic and fulvic acids, pesticides, pathogens, etc. It is a non-specific, but inclusive parameter for monitoring organics. Knowing and understanding TOC levels coming into, throughout, and leaving a plant can be used as a measure of treatment efficacy and as an indicator of contamination. As opposed to methods like BOD and COD, TOC includes all organic compounds and can be achieved in a matter of minutes with instrumentation as opposed to hours or days with reagents in a laboratory.

This paper discusses three examples of municipal drinking water and waste water treatment plants that have implemented online TOC monitoring as a tool to make informative and rapid treatment decisions, allowing them to optimize their plants processes and operations: City of Boulder (75th Street) Public Works Wastewater Treatment Facility, Colorado (USA), Twin Oaks Valley Water Treatment Plant in San Marcos, California (USA) and City of Englewood Water Treatment Plant, Colorado (USA)




## 2 City of Boulder Public Works Wastewater Treatment Facility, Colorado (USA)

### 2.1 Objective

The City of Boulder 75th Street Wastewater Treatment Facility (WWTF), USA gained insight and determined optimization opportunities through the use of online TOC monitoring implemented since March 2015. In addition, the city is looking to

gain approval for long-term BOD:TOC correlations from the State of Colorado in order to replace biochemical oxygen demand (BOD) analysis with TOC analysis which is a faster, easier, and more accurate method of measuring the organic strength of wastewater.

The City of Boulder's (WWTF) 2008 upgrades marked an important transition from a trickling filter/solids contact process to a Modified Ludzack-Ettinger (MLE) biological nutrient removal process. The new activated sludge process has successfully

reduced effluent ammonia and nitrate concentrations to levels comfortably below current Colorado Discharge Permit System (CDPS) discharge permit limits. However, effective December 1, 2017, the same permit proposes lower daily maximum ammonia limits and new daily maximum nitrate limits. If the Boulder WWTF's future nitrate limit (17.9 mg N/L for flows ≥ 20 MGD) were imposed on effluent nitrate quality from 2011 to 2014, 111 violations would appear, illustrating the future vulnerability of the current WWTF configuration.

**2.2 Results**

On site testing and process modeling pointed to the same cause of incomplete denitrification: a carbon limitation in the anoxic zones of the WWTF's MLE process. The use of data from an online TOC Analyzer allowed the City of Boulder WWTF to demonstrate that the diurnal patterns of carbon and nitrogen were offset enough to contribute to the WWTF's carbon limitation. This presents a host of optimization opportunities that were previously overlooked as carbon/nitrogen ratios were considered

on a daily basis. The city's Nitrogen Upgrades Project, currently in the construction phase, will address the WWTF's carbon limitation by implementing external carbon addition via the sugary by-product of the beer brewing process from a nearby brewery and acetic acid.

GE's InnovOx* TOC Analyzer is being used in this study to provide online monitoring of aeration basin influent (ABI) TOC concentrations. The Analyzer collects a sample from a continuously pumped stream and uses heated persulfate oxidation

chemistry assisted by supercritical water to oxidize organic carbon. During this supercritical water oxidation (SCWO), the Analyzer's reactor is heated to 375°C and pressurized to 220bar, which conditions are beyond water's critical point.

After implementing online TOC analysis, The City of Boulder WWTF demonstrated that the diurnal patterns of carbon and nitrogen are offset enough to contribute to the WWTF's carbon limitation. Data showed that the peak nitrogen loading of the plant occurs approximately eight hours before the peak carbon loading. Therefore, the biological denitrification process has

its highest carbon requirement (due to the highest nitrogen input) hours before it actually receives its highest carbon input. This disconnect between nutrient loading and nutrient requirement presents a host of optimization opportunities that were previously overlooked since as carbon:nitrogen ratios were originally determined via a daily composite which masked the actual offset in the timing of the peak load.

Figure 1 (a) and (b) show the diurnal patterns of ammonia and TOC at the aeration basin influent (ABI) and of nitrate at the

secondary clarifier influent (SCT Eff) on weekly and daily cycles. Ammonia and nitrate account for the majority of the inorganic nitrogen in the ABI and the SCT Eff, respectively, so these trends can be approximated to be total nitrogen trends on both the influent and effluent of the activated sludge system. As described previously with an 8-hour delay between the daily nitrogen peak, which occurs in the morning (around 11:30 am) and daily carbon peak, which occurs in the early evening (7-8pm), it is apparent that nitrogen moves through the activated sludge system before peak influent carbon occurs at the

aeration basin influent. This offset in diurnal nitrogen and carbon patterns is a significant contributing factor to the WWTF's carbon limitation.



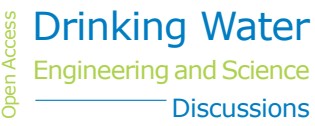

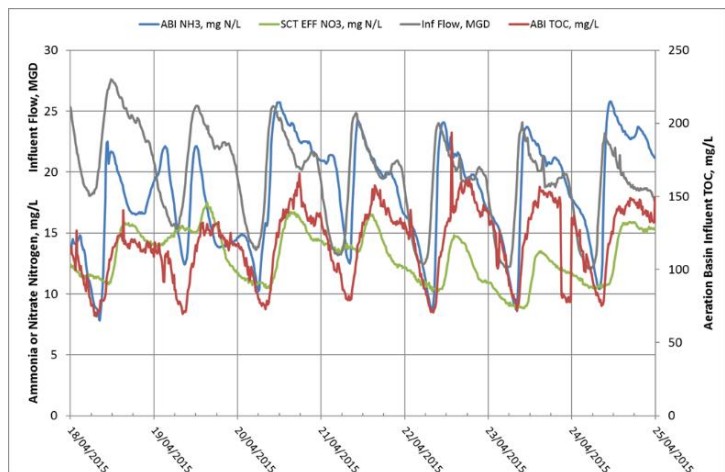

Figure 1 (a): Weekly diurnal patterns of ammonia and TOC at the aeration basin influent (ABI) and of nitrate at the secondary clarifier influent (SCT Eff)

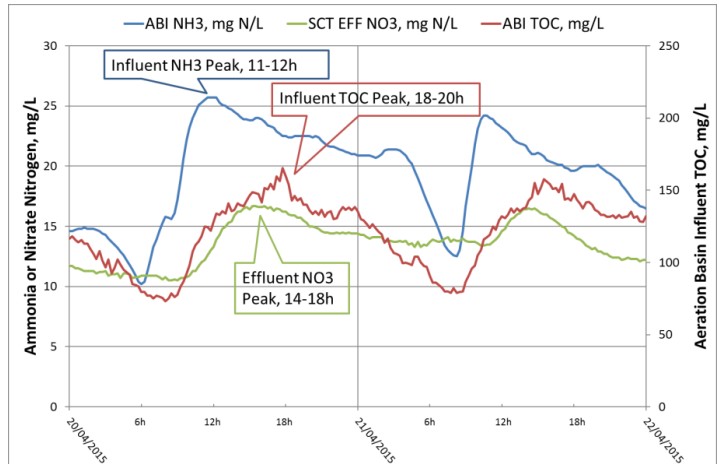

Figure 1 (b): Daily diurnal patterns of ammonia and TOC at the aeration basin influent (ABI) and of nitrate at the secondary clarifier influent (SCT Eff)

To further investigate how the offset of diurnal nitrogen and carbon peaks affects denitrification, a calibrated diurnal model will be developed by the plant's engineers using Dynamita's SUMO process simulation software. Key objectives of the modeling effort will be to:

- Determine the optimum set points for the carbon feed system control philosophy
- Determine how to most efficiently use and control the primary clarifier bypass option
- Adjust the side stream ammonia load to improve the secondary influent C/N ratio

In addition to providing insight into diurnal variability of the WWTF's carbon limitation, TOC is a faster, easier, and more accurate alternative to BOD. Indeed, TOC is a direct measurement of gross amount of organic matter in waters, including suspended particulates, colloidal and dissolved organic matter, while BOD measures the biologically active organic matter indicating amount of oxygen needed for the biological degradation. Every organic compound has a different BOD. Therefore, BOD is poor in precision, and takes 5 days to generates a result, which is not useful for process monitoring. TOC, however, generates a result every few minutes (typically less than 10) and has a more stable baseline.



While BOD and cBOD limits have appeared in NPDES permits since their inception, The Code of Federal Regulations (40CFR133.104(b)), Standard Methods (APHA, 2012), and the EPA's NPDES Permit Writers' Manual (US EPA, 2010) all allow the replacement of BOD methods with TOC methods following the development of long-term

site-specific correlations. The City of Boulder's WWTF engaged in a long term correlation study starting September 2013,
measuring TOC in influent, aeration basin influent, and final effluent using several TOC methodologies on 24-hour flow-based composite samples, that were also analyzed for BOD/cBOD. Regression equations were developed from long-term correlations at each process area according to APHA, 2012 to estimate BOD and cBOD from TOC and are illustrated in Table 1. This data was submitted to the Colorado Water Quality Control Division for approval and inclusion into the city's CDPS discharge permit, which expired April 30, 2016 and, as of the time of publication, is on administrative extension.

Table 1: Summary of the City of Boulder's long-term correlation between BOD and TOC and between cBOD and TOC for both plant influent and final effluent wastewater matrices.

| Wastewater Matrix | Correlation | Number of Data Pairs | Linear Regression Best Fit Equation | $R^2$ |
|---|---|---|---|---|
| Influent | BOD:TOC | 27 | BOD = 1.7607 (TOC) + 13.716 | 0.7123 |
| | cBOD:TOC | 27 | cBOD = 1.2842 (TOC) + 11.184 | 0.6714 |
| Effluent | BOD:TOC | 80 | BOD = 1.8464 (TOC) − 8.241 | 0.5137 |
| | cBOD:TOC | 80 | cBOD = 0.7561 (TOC) +2.5513 | 0.3698 |

With the number of data pairs used for each correlation, the table shows the linear regression best fit line equation and R2 value associated with each correlation.

**3 Twin Oaks Valley Water Treatment Plant in San Marcos, California (USA):**

**3.1 Objective**

The Twin Oaks Valley Water Treatment Plant in San Marcos, CA commissioned in 2008 is a zero discharge plant and one of the world's largest submerged membrane ultrafiltration water treatment plants (100MGD). The plant uses GE Water & Process Technologies ZeeWeed* 1000 ultrafiltration (UF) membranes in its treatment process. The source water is 95% surface water that is mixed with reclaim water on site from an equalization (EQ) basin. The reclaim water is primarily backwash from the
UF membrane trains. The process of recycling water on site starts with equalization followed by addition of coagulant/ flocculant and then settling in Lamella plate settlers. The settled water is combined with the raw water and fed to the UF membranes.

In order to optimize membrane performance, treatment processes and organic loading of the membranes must be monitored closely to minimize organic and inorganic fouling potential.

The purpose of this study was to use online total organic carbon (TOC) monitoring of the influent and effluent to the plate settlers in order to optimize the treatment of recycled water. Near real-time online analysis of the organic carbon removal for different chemical treatments allowed for rapid understanding of the best treatment options and optimization of treatment as shown in Figure 2. For example, in this case, online analysis of the organic carbon contributed to understanding source water better and in real-time so smarter decisions could be made to chemical dosages adjustments, protecting membranes from
fouling (increasing their life time), and finally contributing to saving money on operational expenditures, while making effluent quality better.

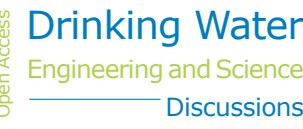

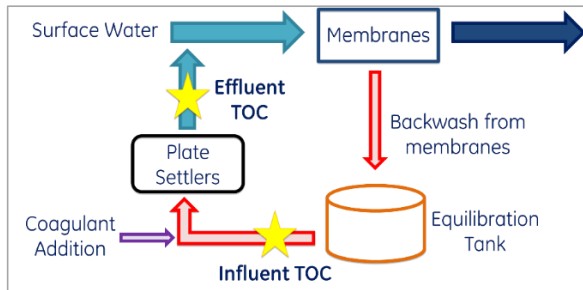

Figure 2: Schematic flow diagram of Twin Oaks Water Treatment Plant's implementation of TOC Analyzer.

### 3.2 Results

Organic carbon levels for the two streams (influent and effluent to the plate settlers) were measured using a Sievers InnovOx
5    On-Line TOC Analyzer as shown in Figure 3. The InnovOx Analyzer uses supercritical water oxidation (SCWO) to oxidize
organics and non-dispersive infrared (NDIR) detection to determine organic carbon concentrations.  The robust oxidation
technology and hardware design provide reliable organic carbon oxidation data even for complex matrices.

For this study, the Analyzer was run in non-purgeable organic carbon (NPOC) mode.  NPOC mode involves acidification of
the sample followed by sparging with CO2-free air in order to remove any inorganic carbon in the sample prior to oxidation.

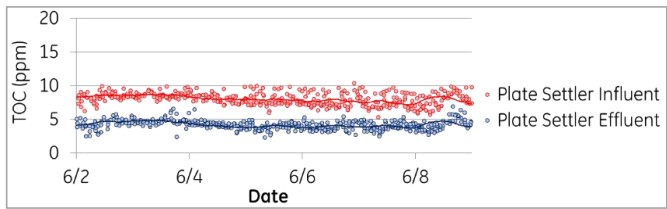

10

Figure 3: TOC for both streams showing removal from influent to effluent

Online analysis of plate settler influent and effluent TOC showed an initial TOC removal efficiency of about 40-50%.  While
trying different chemical treatment options, online TOC analysis provided near real-time insight into the efficiency of the
15    treatment.  Controlling the pH provided better TOC removal efficiency than adding a different coagulant. This is illustrated in
Figure 4.

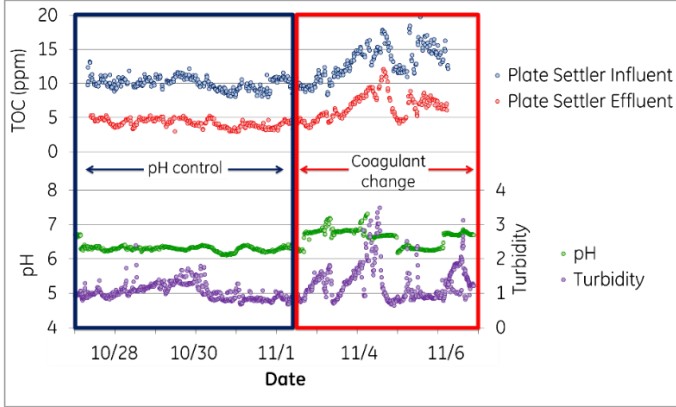

Figure 4: TOC removal efficiencies for pH control and coagulant changes



Future analysis of online TOC for these two streams will continue to provide information on the organic carbon removal efficiency of reclaim water treatment so that membrane performance can be optimized at this plant. Membrane pre-treatment with pH control or coagulant changes can help improve membrane lifetimes, increase backwash cycles, and maintain removal efficiency. If pre-treatment is inadequate it can lead to inorganic fouling (too much coagulant) or organic fouling (too much

organic material). Thus, proper monitoring of organic removal and chemical usage is key to membrane optimization.

As reclaiming and recycling of water becomes increasingly common at industrial and municipal plants, online monitoring of TOC should be used so that water treatment can be optimized for maximum TOC removal.

**4 City of Englewood Water Treatment Plant, Colorado (USA)**

**4.1 Objective**

One of the most valuable ways that TOC analysis can be used in municipal drinking water plants is to understand the amount of disinfection byproduct (DBP) precursors. DBPs form when residual chlorine from disinfection and bromide in water streams react with organic content over time. Known as carcinogens, they are strictly regulated throughout the distribution system. The ultimate dilemma of disinfection is the need to balance disinfectant dosing to control microbial risk with TOC removal to control DBP formation.

Enhanced coagulation is one of the means to decrease TOC content of water. It can be optimized using jar testing as a tool for proactive process control in order to simulate the performance of various chemical coagulants and process conditions without having to test the full-scale treatment process. For many plants, the rule requires optimization of the treatment process to increase the removal of TOC, which can often be improved by selecting the optimum dose of aluminium or ferric-based coagulant. Other treatment parameters including the addition of permanganate, powdered activated carbon, or pH adjustment

can also be easily modelled.

Traditionally, turbidity and UV254 have been used as primary indicators of good floc formation and removal of organics in jar tests. Turbidity is an indicator of water clarity but does not distinguish between inorganic, organic, or particulate contaminant. UV254 measures the aromatic content of organic matter in water, but not all organic molecules absorb in that wavelength and there are multiple interferences at 254nm, such as ferric compounds, which can lead to either over or under

reporting of the estimated organic carbon content of the water.

More recent testing has shown that TOC may be a far better indicator of a fully optimized treatment process. This is particularly true if TOC measurements can be made immediately as various process changes are made to a jar testing plan. Real-world advantages of fully optimized jar tests may include reduced chemical usage or cost, improved removal of organics, minimization of membrane fouling, minimization of sludge production, and a reduction in regulated DBPs. Jar testing is

beneficial for plants so they can optimize their treatment processes to pick the right coagulant type and coagulant dosage.

City of Englewood, CO is a drinking water treatment plant, that treats surface water from South Platte River with a 28 MGD conventional treatment. They were using 60ppm of coagulant (alum sulphate) and expressed desire to reduce chemical costs.

**4.2 Results:**

In order to conduct their process improvement, the City of Englewood expanded their process data for jar testing from just

turbidity to include TOC.

By adding in TOC data to jar testing, the plant was able to save over $100k in chemicals and disposal costs and shown in Table 2. They also realized that effective TOC removal does not always correlate to effective turbidity removal or vice versa, therefore TOC and turbidity levels must both be monitored. Typical coagulants can remove TOC to a certain degree, beyond that amount excess chemical is a waste of money and requires excess sludge removal. Characteristics of a plant's source water

can change rapidly, including pH, alkalinity, and the organic composition of the water. On-line TOC monitoring is the most effective means for frequent process observation.





| | Dosage (mg/l) | Coagulant Usage /year | Coagulant Costs /year | Coagulant Savings /year | Coagulant Waste /year | Disposal Costs /year | Total Savings /year |
|---|---|---|---|---|---|---|---|
| Stage 1: D/DBPR implemented | 60 | 1 410 588 lbs | $ 136 827 | NA | 1 830 yards$^3$ | $ 100 650 | NA |
| Coagulant reduction | 45 | 959 049 lbs | $ 106 454 | $ 30 373 | 1 250 yards$^3$ | $ 68 750 | $ 62 723 |
| 1st optimization study with TOC | 36 | 728 028 lbs | $ 86 003 | $ 50 824 | 920 yards$^3$ | $ 50 600 | $ 100 874 |
| 2nd optimization study with TOC | 20 | 426 174* | $ 53 698* | $ 83 129* | 700 yards$^{3*}$ | $ 38 500* | $ 145,279* |

*Usage, costs and savings are calculated for one year based on current dosage rate recently implemented

Table 2: Chemical and disposal cost savings achieved by adding in TOC analysis

Before conducting any process improvements trials, they were dosing chemicals without any controls in place. With the new DBP regulations in place, it was estimated easier to dose excessive chemicals and meet the regulations requirements, than not doing it. This resulted in very high sludge production and costly sludge removal. They managed to reduce operational cost expenditures within several steps of plant optimization, including the ability to change pH, coagulant type, or coagulant dosage to obtain optimum results and ensure removal of organics and know when to regenerate granular activated carbon (GAC).

**5 Conclusions:**

Online organic carbon monitoring drives smart, informative and rapid decision making to improve process control of drinking water and wastewater treatment plants so that these treatment facilities can meet regulatory compliances and/or optimize treatment process. Municipal treatment facility operators can use data to make real time actions that impact their OPEX spending and their capabilities to meet regulatory requirements.

These three examples of plants demonstrated that the use of data from a TOC Analyzer provides insights of real-time variations of organic carbon, that can be used to optimize processes, ranging from nutrient dosing at a biological wastewater treatment facility to treating membrane backwash water to minimizing DBP formation potential in drinking water.

Implementing TOC analysis at water treatment facilities is a powerful tool that can help operators continue to effectively treat water and positively impact the costs of treatment, in order to meet current and future regulatory requirements.

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

Dondra Biller; Mark Mullet (2016): Optimizing Treatment of Reclaimed Water at a Drinking Water Plant by Online Monitoring of Organic Carbon Levels. Pittcon 2016, Atlanta, GO

Cole Sigmon, Melissa Mimna, Leah Santiago, and Mark Mullet (2016): TOC Talks: Insight and Efficiency at the City of Boulder's Wastewater Treatment Facility. WEFTEC 2016, New Orleans, LA

*A trademark of General Electric Company. May be registered in one or more countries.