# Peer review of "Online Total Organic Carbon (TOC) monitoring for water and wastewater treatment plants processes and operations optimization"

_Drinking Water Engineering and Science, 2017_

## Referee Comment (RC1) · Anonymous Referee #1 · 31 Mar 2017

The paper as presented showed some interesting examples of the use of TOC monitoring but was missing quite a bit. In the abstract it talks about BOD & COD not being useful for process control which is true which was applicable to one of the case studies demonstrated but not to the other two. The paper also seemed overly biased to one particular product in general and seemed more of a marketing paper rather than looking at the benefits of the technique as a whole.

It was very disappointing that in the references that there was no detailed reference to the work that WEF members have done on using TOC as an alternative to BOD for regulatory purposes as this was a seminal moment for the use of the technique this seems like quite a big thing to be missing in the paper.

[Figure]

The paper is also a bit sensationalist digging into the BOD and COD techniques whilst not admitting the drawbacks of TOC (i.e. the cost and complexity of the technique).

I do think that the paper provides value which is why i haven't rejected it outright but I think it needs a major re-write with a big adjustment in the balance of the paper with a little bit more research into the usefulness of the analytical technique in the industry rather than being focused on one particular instrument and its uses

---

## Referee Comment (RC2) · Anonymous Referee #2 · 4 Apr 2017

1. Does the paper address relevant scientific questions within the scope of DWES?

   *This paper addresses a relevant and interesting topic for DWES. Especially for the special CCWI 2016 issue, the application of online analyzers for advanced control and monitoring fits the scope.*

2. Does the paper present novel concepts, ideas, tools, or data?

   *This paper presents a further refinement of the application of online analyzers for advanced control and monitoring. The application of online TOC measurements for advanced control for waste water treatment plants in the Netherlands is not yet widely spread. Especially the idea of applying online TOC measurements optimizing the amount of BOD available for the denitrification process in a waste*

[Figure]

*water treatment plant is interesting. (As an addition to ammonium and nitrate analyzers)*

3. Are substantial conclusions reached?

   *Based on the results conclusions can be reached; however the conclusions could be more specifically formulated. In the current version they are quite qualitative and not quantitative. For example: due to optimization with TOC, the original dosage can be reduced to approximately one third.*

4. Are the scientific methods and assumptions valid and clearly outlined?

   *Improvements can be made for this aspect. First of all, the method for each case is not structured in a distinct subsection (further elaborated at aspect no. 10.)*

   *The method for each section is clearly explained, but it is not compared with methods from other studies. Also explicit references to other works are not present in these paragraphs. (See also aspect no. 7.)*

5. Are the results sufficient to support the interpretations and conclusions?

   *The reason for a selected period of the results shown should be made clear. Are these results a specific selection of a period or are these the results of a whole dataset? Also here, just like with the methods, a critical comparison with the results of previous or other studies is not made. (See also aspect no. 7.)*

6. Is the description of experiments and calculations sufficiently complete and precise to allow their reproduction by fellow scientists (traceability of results)?

   *The applied methods and measurement instruments are sufficiently described.*

7. Do the authors give proper credit to related work and clearly indicate their own new/original contribution?

*No. This is the major improvement to be made in this paper. No citations or references to related work are made in the text. Also a critical comparison of the results with other works is not made. It is advised to rewrite this paper in such a way that also the broader discussion on this topic becomes clear and also the position of this paper in this discussion.*

8. Does the title clearly reflect the contents of the paper?

   *Yes. It clearly reflects the contents of the paper.*

9. Does the abstract provide a concise and complete summary?

   *Yes. It covers the contents of the paper.*

10. Is the overall presentation well structured and clear?

    *The main structure of the paper is clear. However, the structure for each case can be improved. In the current structure, there is a distiction between objective and results. However, in the subsection about results, also the methods are included. For each case a clear distinction between methods and results should be made.*

    *Section 2: The first two paragraphs of 2.2 are not results but methods. I would suggest to make an extra subsection 2.2 Methods, and subsection 2.3 with Results, and subsection 2.4 with Outlook (or futher investigation).*

    *Section 3: In this section the distinction between objective, methods, results and outlook is not clear. Also here I would like to suggest to make a strict distinction between these subsections*

    *Section 4: Idem dito.*

11. Is the language fluent and precise?

    *Yes. The language is fluent and precise.*

**DWESD**

12. Are mathematical formulae, symbols, abbreviations, and units correctly defined and used?

    *The following abbreviations are not further defined: abstract: OPEX; section 2.2: SUMO.*

13. Should any parts of the paper (text, formulae, figures, tables) be clarified, reduced, combined, or eliminated?

    *The captions of the tables should be placed on top of the tables and not below the tables.*

14. Are the number and quality of references appropriate?

    *The number of references is too low and should be increased. Even more important, there are not any explicit citations or references within in text. There is only a References section included at the end.*

15. Is the amount and quality of supplementary material appropriate?

    *Not available.*

---

## Author Comment (AC1) · 24 Apr 2017

The paper will be reviewed and amended to take referee's comments into consideration, to balance the content toward usefulness in the industry:

- Present the benefits of TOC technology as a whole, Focus less on a particular product

- Include section about the drawbacks of TOC technology, such as cost of ownership and complexity of the technique

- Include the reference of the WEF members work on using TOC as an alternative for BOD, for regulatory purposes

[Figure]

- Include comparison with other methods from other studies, and add explicit references (+ include them in text)

- Include quantification of conclusion, where apply, or formulate better.

- change the structure with clear separation of method and results and outlook (further investigation)

- define SUMO and OPEX abbreviations

- place captions of tables on top

---

## Author Comment (AC2) · 2 May 2017

**References of bibliography for DWES Publication:**

**Background information:**

Biochemical and chemical oxygen demand measurements have been used for over 100 years to qualify and quantify contamination in municipal and industrial wastewater. Biochemical Oxygen Demand, currently a five-day laboratory test labeled $BOD_5$, is one of the most broadly used parameters for wastewater quality in the world and the standard for municipal sewage treatment. Chemical Oxygen Demand (COD), typically a two-hour test, is more widely used in industrial applications. Often, both of these laboratory methods are measured, recorded and compared over time.

Bibliography:

"Pacific Southwest, Region 9 – Quality Assurance." United States Environmental Protection Agency. Last updated 20 May 2016. 24 Aug 2016 www.epa.gov/region9/qa/

"BIOCHEMICAL OXYGEN DEMAND (BOD) – Standard Method 5210 B (5-day BOD Test)." United States Environmental Protection Agency. Last updated 20 May 2016. 24 Aug 2016 www.epa.gov/region9/qa/pdfs/5210dqi.pdf

ASTM D1252 – 06(2012) Standard Test Methods for Chemical Oxygen Demand (Dichromate Oxygen Demand) of Water

**Regulatory framework for the use of COD vs TOC:**

- The COD usual method (DIN 38409-H41) is using Chromate and Mercury, which are very toxic chemicals. For this reason, there is a tendency to move away from the parameter COD and to promote the use of the parameter TOC.
- Some countries, like Sweden, even banned the use of this methodology.
- The development of TOC as a parameter is being reflected in a number of documents, within the Directive Monitoring of emissions to air and water - Industrial Emissions Directive 2010/75/EU (Integrated Pollution Prevention and Control) such as:
    - ROM, final draft document: "Total organic carbon (TOC)/Chemical oxygen demand (COD): In some Member States, there is a trend to replace COD by TOC for economic and environmental reasons. The use of chromate and mercury, necessary for the COD determination, can be avoided by determining TOC, which can be measured continuously by on-line analysers."
    - CWW, final draft document: 'Either TOC or COD is monitored. TOC monitoring is the preferred option, because it does not rely on the use of very toxic compounds.'

Bibliography:

http://eippcb.jrc.ec.europa.eu/reference/BREF/ROM_FD_102013_online.pdf) : Section 4.3.1, page 82
http://eippcb.jrc.ec.europa.eu/reference/BREF/CWW_Final_Draft_07_2014.pdf) : Section 4.2, BAT 4, page 553

**Regulatory framework for rejects monitoring:**

In the United States, pre-treatment standards are established for all industrial and Publicly Owned Treatment Works (POTWs). Under the authority of the Clean Water Act and subsequent legislation, the National Pollutant Discharge Elimination System (NPDES) was established under the administration of the Environmental Protection Agency (EPA). With minimal exceptions, NPDES is the primary program that manages discharge limits or effluent limitations guidelines (ELG) for the release of process effluent or wastewater to public waterways.

Bibliography:

"NPDES Permit Program Basics." United States Environmental Protection Agency. 24 Aug 2016 https://cfpub.epa.gov/npdes/docs.cfm?document_type_id=8&view=Permit%20Applications%20and%20Forms&program_id=45&sort=name

"State NPDES Program Authority." United States Environmental Protection Agency. Last updated 19 Feb 2016. 24 Aug 2016 https://www.epa.gov/npdes/npdes-state-program-information

In Europe, France has effluent discharge limitations in open waterways requiring BOD < 100 mg/L and COD < 300 mg/L . Germany allows a maximum COD value based on 4 x TOC – "a chemical oxygen demand (COD) level specified in the water discharge permit shall also be deemed to have been met provided the quadruple amount of total organically bonded carbon (TOC), specified in milligrams per litre, does not exceed this level."

Bibliography:
"Arrêté du 26 mars 2012 relatif aux prescriptions générales applicables aux installations classées relevant du régime de l'enregistrement au titre de la rubrique n° 2710-2 (installations de collecte de déchets non dangereux apportés par leur producteur initial) de la nomenclature des installations classées pour la protection de l'environnement." Article 35: Valeurs limites de rejet. Current as of 26 March 2012. 21 April 2015 http://legifrance.gouv.fr/eli/arrete/2012/3/26/DEVP1208907A/jo/ar- ticle_35

"Promulgation of the New Version of the Ordinance on Requirements for the Discharge of Waste Water into Waters." Article 6, section (3), page 5. Federal Ministry for the Environment, Nature Conservation and Nuclear Safety, Germany. Current as of 17 June 2004. 24 Aug 2016 http://www.bmub.bund.de/fileadmin/bmu-import/files/pdfs/allgemein/application/pdf/wastewater_ordinance.pdf

**Value of TOC to Oxygen Demand Correlation**

TOC analysis is faster and more accurate than either oxygen demand method and is a direct measurement of the organic load. Both forms of oxygen demand are indirect measurements. TOC has an analysis time of 3 to 10 minutes, or 30 minutes for at least three (3) repetitions, compared to two hours for COD or five days for $BOD_5$.

The NPDES system allows for "authorized alternatives" to oxygen demand, such as TOC measurement, correlating to oxygen demand, as a means for operators to have faster and more accurate monitoring and process control. In this way, industrial facilities, "non-municipal dischargers", with wastewater treatment can often trend oxygen demand and anticipate excursions before exceeding their permit limits.

Bibliography:
"Central Tenets of the National Pollutant Discharge Elimination System (NPDES) Permitting Program." Page 2. United States Environmental Protection Agency. Last updated 7 April 2015. 21 April 2015 http://water.epa.gov/polwaste/npdes/basics/upload/tenets.pdf

A pre-treatment facility should work with its state NPDES administrator to execute a long-term, correlation test and replace BOD or COD with TOC as the primary discharge parameter. Regulatory agencies (e.g., USEPA, state DEPs) may have specific requirements on the number of samples and test period. As indicated in a study report by Instrumentation Testing Association of North America, "weekly sample analysis for a minimum of one year to include seasonal variations is recommended for municipal wastewater plant in order to obtain discharge permit".

Bibliography:
Nutt, Stephen G. and Tran, John of XCG Consultants Ltd. "Addressing BOD5 limitations through Total Organic Carbon Correlations: A Five Facility International Investigation." Pensacola, Florida: water & Wastewater Instrumentation Testing Association of North America (ITA). January 2013.

Since TOC and oxygen demand methods are inherently different, the historical concern with TOC:COD correlation is the stability of the relationship over time due to any changes in the process stream(s). The variability of organics over time could alter the mathematical relationship to oxygen demand. The

sample matrix, particulate or solids composition, viscosity and turbidity can influence the correlation factor over time.

By measuring TOC every ten minutes and applying the correlation factor:

- COD can be estimated as many as 12 times more frequently than the traditional test
- $BOD_5$ can be estimated 288 times per day, compared to the traditional test

In Ireland, Inflluent and effluent samples from 11 WW treatment plants were analyzed.

Result are: Influent agreeable for both BOD and COD to be replaced by TOC. Effluent relationship between COD and TOC but not BOD.

Bibliography:

Replacement of chemical oxygen demand (COD) with total organic carbon (TOC) for monitoring wastewater treatment performance to minimize disposal of toxic analytical waste

Authors: Donata Dubbera; Nicholas F. Graya

Affiliation: Water Technology Research Group, Centre for the Environment, School of Natural Sciences, Trinity College Dublin, Dublin, Ireland

Conclusion that for treatment performance monitoring COD can be reliably replaced with the TOC. Predictive equations could be developed to estimate COD from TOC measurements.

Bibliography:

Replacement of chemical oxygen demand (COD) with total organic carbon (TOC) for monitoring wastewater treatment performance to minimize disposal of toxic analytical waste

Authors: Donata Dubbera; Nicholas F. Graya

Affiliation: Water Technology Research Group, Centre for the Environment, School of Natural Sciences, Trinity College Dublin, Dublin, Ireland

**How to Determine the Correlation Factor**

There are a number of ways to properly determine the correlation factor between TOC and the oxygen demand parameter of choice, $BOD_5$ or COD. The method detailed in the Instrumentation Testing Association (ITA) Test Report is specific with corresponding statistical analyses; refer to the Implementation Protocol.

Bibliography:

Nutt, Stephen G. and Tran, John of XCG Consultants Ltd. "Addressing BOD5 limitations through Total Organic Carbon Correlations: A Five Facility International Investigation." Pensacola, Florida: water & Wastewater Instrumentation Testing Association of North America (ITA). January 2013.

---

## Author Comment (AC3) · 25 May 2017

The paper was reviewed and amended to take referee's comments into consideration, to balance the content toward usefulness in the industry: - Present the benefits of TOC technology as a whole, Focus less on a particular product - Include section about the drawbacks of TOC technology, such as cost of ownership and complexity of the technique - Include technical advantages and drwbacks of each TOC, COD and BOD technologies - Refer to other publications supporting as well that TOC monitoring is more appropriate for process control than COD or BOD (reference # 1 and 2) - the WEF work for alternatives to BOD are now considered and referenced in 8 and 9

The rewrite is now available for your review: (revision 3). And highlight corrections is

attached as a file to this reply. thank you Céline Assmann

Please also note the supplement to this comment:
http://www.drink-water-eng-sci-discuss.net/dwes-2017-11/dwes-2017-11-AC3-supplement.pdf

---

## Author Comment (AC4) · 25 May 2017

**Point by point response to the reviews & list of changes made to Manuscript Correction 2 from May 2017**

**"Online Total Organic Carbon (TOC) monitoring for water and wastewater treatment plants processes and operations optimization"**
**by Céline Assmann et al.**

**Céline Assmann et al.**
celine.assmann@ge.com

The paper was reviewed and amended to take referee's comments into consideration, to balance the content toward usefulness in the industry:
- Present the benefits of TOC technology as a whole, Focus less on a particular product
- Include section about the drawbacks of TOC technology, such as cost of ownership and complexity of the technique
- Include the reference of the WEF members work on using TOC as an alternative for BOD, for regulatory purposes
- Include comparison with other methods from other studies, and add explicit references (+ include them in text)
- Include quantification of conclusion, where apply, or formulate better.
- change the structure with clear separation of method and results and outlook (further investigation)
- define OPEX abbreviation
- place captions of tables on top

**Online Total Organic Carbon (TOC) monitoring for water and wastewater treatment plants processes and operations optimization**

Céline Assmann[1], Amanda Scott[1], Dondra Biller[1]

[1]Analytical Instruments, a Division of GE Power, Boulder, Colorado, USA

5  *Correspondence to*: Céline Assmann, celine.assmann@ge.com; Amanda Scott, amanda.scott@ge.com

**Abstract.** Organic measurements, such as biological oxygen demand (BOD) and chemical oxygen demand (COD) were developed decades ago in order to measure organics in water. Today, these time-consuming measurements are still used as parameters to check the water treatment quality; however, the time required to generate a result, ranging from hours to days, does not allow COD or BOD to be useful process control parameters [1],[2]. Online Organic Carbon Monitoring allows for

10  effective process control because results are generated every few minutes. Though it does not replace BOD or COD measurements still required for compliance reporting, it allows smart, data-driven and rapid decision-making to improve process control and optimization or meet compliances. Thanks to the smart interpretation of generated data and the capability to now take real-time actions, municipal drinking water and wastewater treatment facility operators can positively impact their OPEX (Operational Expenditure) efficiencies and their capabilities to meet regulatory requirements. This paper describes how

15  three municipal wastewater and drinking water plants gained process insights, and determined optimization opportunities thanks to the implementation of on-line TOC monitoring.

**1 Introduction**

Growing populations and expanding industries are pulling on water resources while adding nutrients and pollutants to water sources. These facts coupled with heightened public demand for quality water at affordable prices has the water industry under

20  scrutiny. Whether complying with water regulations, optimizing treatment processes for saving time and money, or looking to better manage a plant during times of emergency (flood, fire, security threat, drought or industrial spill), knowing and understanding organics and organic removal can be extremely valuable. Total Organic Carbon (TOC) Monitoring is one of the most important parameters that drinking water and wastewater facilities can use to make decisions about treatment.

Measuring TOC can be critical to a water treatment facility's water quality in helping to optimize treatment processes. TOC is

25  useful in detecting the presence of many organic contaminants including petroleum products, organic acids like humic and fulvic acids, pesticides, pathogens, etc. It is a non-specific, but inclusive parameter for monitoring organics. Knowing and understanding TOC levels coming into, throughout, and leaving a plant can be used as a measure of treatment efficacy and as an indicator of contamination. As opposed to methods like BOD and COD, TOC includes all organic compounds and can be achieved in a matter of minutes with instrumentation as opposed to hours or days with reagents in a laboratory.

30  This paper discusses the three organics measurement methodologies mostly used today (BOD, COD,TOC) and provides examples of three municipal drinking water and waste water treatment plants that have implemented online TOC monitoring as a tool to make informative and rapid treatment decisions, allowing them to optimize their plants processes and operations: City of Boulder (75th Street) Public Works Wastewater Treatment Facility, Colorado (USA), Twin Oaks Valley Water Treatment Plant in San Marcos, California (USA) and City of Englewood Water Treatment Plant, Colorado (USA).

35

**2 Discussion of the methods for organics measurements and regulatory frameworks**

**2.1 The methods for organics measurements in water and wastewater**

Since the 1970s, laboratory analytical methods for organics measurements have been developed with the aim to establish the concentration (typically in mg/L or ppm) of organics (i.e., carbon-containing) matter to determine the relative "strength" of a water and a wastewater sample. Today there are three common laboratory tests used to determine the gross amount of organic matter: BOD (biochemical oxygen demand), COD (chemical oxygen demand) and TOC (total organic carbon). Though these tests measure different things in water, there is overlap in the results, and some correlations could be established [15].

**2.1.1 BOD measurements**

BOD measures the amount of dissolved oxygen needed by aerobic biological organisms to oxidise organic material in a water sample. BOD is commonly expressed as BOD5, mg of $O_2$ consumed per litre of sample during 5 days of incubation at 20°C. It is an indirect measurement of organic quality or pollution in water [1].

cBOD (Carbonaceous BOD) is a BOD measurement where a nitrification inhibitor is added to the BOD sample, to stop the oxidation of ammonia to nitrate, and measure specifically the organic carbon contribution to oxygen demand.

To ensure proper biological activity during the BOD test, a water sample must be free of Chlorine and Copper, in pH range 6.5 to 7.5, and needs to have adequate microbiological population. Besides this, BOD test is well known to have a challenging reproducibility from person to person, and generates a result after the 5 days of incubation.

**2.1.2 COD measurements**

COD is a popular alternative and complementary test to BOD, with the major advantage that it only takes few hours to complete, compared to the 5 days for BOD. COD analysis is based on the principle to measure the change in colour caused by the chemical oxidation of the sample. The oxidation is achieved by closed reflux of a potassium dichromate in sulfuric acid solution. Similarly to BOD Analysis, it is an indirect measurement of organic quality or pollution in water and is commonly expressed as mg of $O_2$ consumed per litre of sample [2].

COD analysis uses toxic chemicals and generates hazardous waste, that require proper handling and disposal. Indeed, along with the potassium dichromate in 50% sulfuric acid solution, pre-prepared COD vials also contain silver sulfate as a catalyst and mercuric sulfate to mitigate chloride interferences.

**2.1.3 TOC measurements**

The TOC test is gaining popularity because it only takes 5-10 minutes to complete. At the heart of the TOC test is a carbon analyzing instrument that measures the total organic carbon in a water or wastewater sample. There are different types of analyzers, but all oxidize organic carbon into carbon dioxide ($CO_2$) and measure that $CO_2$ generated using a detection method. Oxidation methods include combustion, UV persulfate, and Super Critical Water Oxidation while detection methods include NDIR (non-dispersive infrared) and membrane conductivity [17],[18].

COD and BOD are laboratory techniques whereas TOC can be done in the laboratory (off-line measurements) or online (at-line measurements). The value of online analysis is obviously getting real time data to see process changes and make quick process decisions based on the observed fluctuations. Online TOC analyzers typically require maintenance throughout the year and have consumable parts that need to be changed out. Newer TOC analyzers however are designed for ease of use and have minimized maintenance down to once per quarter with calibration every 6-12 months.

The cost of ownership and complexity is more important with TOC than with COD or BOD: TOC test procedures are relatively simple and straight-forward, but are specific to the type of carbon-analyzing instrument utilized. Thus, no "typical" TOC procedure exists. The instrument manufacturer's procedures should be followed accurately to achieve the best results.

TOC is a highly sensitive, non-specific measurement of the organics present in a sample. Suspended particulate, colloidal, and dissolved organic matter are part of the TOC measurement.

**2.2 Regulatory frameworks**

The COD usual method (DIN 38409-H41) is using Chromate and Mercury, which are toxic chemicals. For this reason, there is a tendency to look for alternatives to the parameter COD and to promote the use of the parameter TOC or Chrom-Free COD.

In Europe, the development of TOC as a parameter is being reflected in a number of documents, within the Industrial Emissions Directive 2010/75/EU (Integrated Pollution Prevention and Control) such as ROM (Report On Monitoring of Emissions from IED-Installations), final draft document: "Total organic carbon (TOC)/Chemical oxygen demand (COD): In some Member States, there is a trend to replace COD by TOC for economic and environmental reasons. The use of chromate and mercury, necessary for the COD determination, can be avoided by determining TOC, which can be measured continuously by on-line analysers." [4]. Some countries, like Sweden, are looking for alternative technologies [3].

In the USA, National Pollutant Discharge Elimination System (NPDES) was established under the administration of the Environmental Protection Agency (EPA). With minimal exceptions, NPDES is the primary program that manages discharge limits or effluent limitations guidelines (ELG) for the release of process effluent or wastewater to public waterways [5],[6]. The NPDES system allows for "authorized alternatives" to oxygen demand, such as TOC measurement, correlating to oxygen demand, as a means for operators to have faster and more accurate monitoring and process control [7].

**2.3 Discussion about the determination of the correlation factor**

There are a number of ways to properly determine the correlation factor between TOC and the oxygen demand parameter of choice, BOD5 or COD. The method detailed in the Instrumentation Testing Association (ITA) Test Report is specific with corresponding statistical analyses; refer to the Implementation Protocol [8].

A treatment facility should work with its state NPDES (or other local authority in other countries, like DREAL in France) administrator to execute a long-term, correlation test and replace BOD or COD with TOC as the primary discharge parameter. National regulatory agencies (e.g., USEPA, state DEPs in USA) may have specific requirements on the number of samples and test period [8].

[revised manuscript text omitted]

In order to conduct their process improvement and find cost savings opportunities, the City of Englewood expanded their process data for jar testing from just turbidity to include TOC. Before conducting any trials, they were dosing chemicals blindly to ensure compliance with the new DBP regulations which require both TOC removal and minimizing formation of DBPs at the furthest point in their distribution system. By dosing excess chemicals, they were able to meet regulations but this also led to high chemical costs, high sludge production, and costly sludge removal.

**5.2 Results and further investigation**

They managed to reduce operational cost expenditures within several steps of plant optimization, including the ability to change pH, coagulant type, or coagulant dosage to obtain optimum results and ensure removal of organics and know when to regenerate granular activated carbon (GAC).

By having TOC analysis on-site and jar testing data with TOC and turbidity, plant operators did not have to wait for third party test results and could make immediate process decisions.

The plant was able to save over $100k in chemicals and disposal costs and shown in Table 2. They also realized that effective TOC removal does not always correlate to effective turbidity removal or vice versa, therefore TOC and turbidity levels must both be monitored. Typical coagulants can remove TOC to a certain degree, beyond that amount excess chemical is a waste of money and requires excess sludge removal. Characteristics of a plant's source water can change rapidly, including pH, alkalinity, and the organic composition of the water. On-line TOC monitoring is the most effective means for frequent process observation.

Table 2: Chemical and disposal cost savings achieved by adding in TOC analysis

| | Dosage (mg/l) | Coagulant Usage /year | Coagulant Costs /year | Coagulant Savings /year | Coagulant Waste /year | Disposal Costs /year | Total Savings /year |
|---|---|---|---|---|---|---|---|
| Stage 1: D/DBPR implemented | 60 | 1 410 588 lbs | $ 136 827 | NA | 1 830 yards$^3$ | $ 100 650 | NA |
| Coagulant reduction | 45 | 959 049 lbs | $ 106 454 | $ 30 373 | 1 250 yards$^3$ | $ 68 750 | $ 62 723 |
| 1$^{st}$ optimization study with TOC | 36 | 728 028 lbs | $ 86 003 | $ 50 824 | 920 yards$^3$ | $ 50 600 | $ 100 874 |
| 2$^{nd}$ optimization study with TOC | 20 | 426 174* | $ 53 698* | $ 83 129* | 700 yards$^{3*}$ | $ 38 500* | $ 145,279* |

*Usage, costs and savings are calculated for one year based on current dosage rate recently implemented

Further investigation consists in using TOC data and TOC characterization to try and better understand what types of organics are impacting treatment such as coagulant dose, DBP formation, and membrane fouling. Also, a better understanding of source water characteristics and organic loading can help size system processes. As water reuse systems become more viable, TOC analysis gains interest as an indicator for the health of each train in a multiple barrier treatment process, helping both to protect human and environmental health.

**6 Conclusions**

Online organic carbon monitoring drives smart, informative and rapid decision making to improve process control of drinking water and wastewater treatment plants so that these treatment facilities can meet regulatory compliances and/or optimize treatment process. Municipal treatment facility operators can use data to make real time actions that impact their OPEX spending and their capabilities to meet regulatory requirements.

These three examples of plants demonstrated that the use of data from a TOC Analyzer provides insights of real-time variations of organic carbon, that can be used to optimize processes, ranging from nutrient dosing at a biological wastewater treatment facility to treating membrane backwash water to minimizing DBP formation potential in drinking water.

Implementing TOC analysis at water treatment facilities is a powerful tool that can help operators continue to effectively treat water and positively impact the costs of treatment, in order to meet current and future regulatory requirements.

*A trademark of General Electric Company. May be registered in one or more countries.

---

## Author Comment (AC5) · 22 Jun 2017

First, the authors would like to thank the reviewer for taking valuable time to review and for the critical assessment of the paper.

**Comment 1:**
The paper as presented showed some interesting examples of the use of TOC monitoring but was missing quite a bit. In the abstract it talks about BOD & COD not being useful for process control which is true which was applicable to one of the case studies demonstrated but not to the other two.

**Answer 1:**
In the cases presented, TOC measurements are leveraged as a direct or as an indirect tool for processes controls and optimization. Case 1 is an example of direct use, case 2 and 3 are indirect:
- Case 2: in this case, online analysis of the organic carbon contributed to understanding source water better and in real-time so smarter decisions could be made to chemical dosages adjustments. Consequences are a help for operators to protect membranes from fouling while generating better effluent quality.
- Case 3: By having TOC analysis on-site and jar testing data with TOC and turbidity, plant operators did not have to wait for third party test results and could make immediate process decisions: Plant operators managed to reduce operational cost expenditures within several steps of plant optimization, including the ability to change pH, coagulant type, or coagulant dosage to obtain optimum results and ensure removal of organics and know when to regenerate granular activated carbon (GAC).

**Comment 2:**
The paper also seemed overly biased to one particular product in general and seemed more of a marketing paper rather than looking at the benefits of the technique as a whole.

**Answer 2:**
The structure of the paper was revamped to address that comment. Firstly, a precise definition of each organics measurement technology was provided, together with a brief explanation of pros and cons.
The following sections were added:
2 Discussion of the methods for organics measurements and regulatory frameworks
2.1 The methods for organics measurements in water and wastewater
2.1.1 BOD measurements
2.1.2 COD measurements
2.1.3 TOC measurements
2.2 Regulatory frameworks
2.3 Discussion about the determination of the correlation factor

Secondly, the paper is now quoting the particular product or technology used to conduct the study, with less emphasis on it. It was important to explain the technology used, as TOC analyzers have a wide range of applications and technologies, and not all are applicable to measure TOC in the cases illustrated here. We also had to explain what mode was used (NPOC vs TOC) because it has an influence on the results.

**Comment 3:**
It was very disappointing that in the references that there was no detailed reference to the work that WEF members have done on using TOC as an alternative to BOD for regulatory purposes as this was a seminal moment for the use of the technique this seems like quite a big thing to be missing in the paper.

**Answer 3:**
The revised version of the paper includes references to the ITA and WEF work about using TOC as an alternative to COD and BOD, and methodologies to achieve a correlation that can be accepted by national authorities. See especially sections 2.2 and 2.3.

**Comment 4:**
The paper is also a bit sensationalist digging into the BOD and COD techniques whilst not admitting the drawbacks of TOC (i.e. the cost and complexity of the technique).

**Answer 4:**
The TOC monitoring technique has been explained in section 2.1.3, and additional wording about drawbacks added:
The value of online analysis is obviously getting real time data to see process changes and make quick process decisions based on the observed fluctuations. Online TOC analyzers typically require maintenance throughout the year and have consumable parts that need to be changed out. Newer TOC analyzers however are designed for ease of use and have minimized maintenance down to once per quarter with calibration every 6-12 months.
The cost of ownership and complexity is more important with TOC than with COD or BOD: TOC test procedures are relatively simple and straight-forward, but are specific to the type of carbon-analyzing instrument utilized. Thus, no "typical" TOC procedure exists. The instrument manufacturer's procedures should be followed accurately to achieve the best results.

**Comment 5:**
I do think that the paper provides value which is why i haven't rejected it outright but I think it needs a major re-write with a big adjustment in the balance of the paper with a little bit more research into the usefulness of the analytical technique in the industry rather than being focused on one particular instrument and its uses.

**Answer 5:**

The paper was reviewed and amended to take referee's comments into consideration, to balance the content toward usefulness in the industry:

- Present the benefits of TOC technology as a whole, Focus less on a particular product
- Include section about the drawbacks of TOC technology, such as cost of ownership and complexity of the technique
- Include comparison with other methods from other studies, and add explicit references (+ include them in text)

---

## Author Comment (AC6) · 22 Jun 2017

[Figure]

We hope the manuscript will meet your expectations,

Kind Regards,

Céline Assmann

Please also note the supplement to this comment:
http://www.drink-water-eng-sci-discuss.net/dwes-2017-11/dwes-2017-11-AC6-supplement.pdf
* * *
[Figure]

**Supplement:**

First, the authors would like to thank the reviewer for taking valuable time to review and for the critical assessment of the paper.

**Comment 1:**
Does the paper address relevant scientific questions within the scope of DWES?
This paper addresses a relevant and interesting topic for DWES. Especially for the special CCWI 2016 issue, the application of online analyzers for advanced control and monitoring fits the scope.

**Answer 1:**
None

**Comment 2:**
Does the paper present novel concepts, ideas, tools, or data?
This paper presents a further refinement of the application of online analyzers for advanced control and monitoring. The application of online TOC measurements for advanced control for waste water treatment plants in the Netherlands is not yet widely spread. Especially the idea of applying online TOC measurements optimizing the amount of BOD available for the denitrification process in a waste water treatment plant is interesting. (As an addition to ammonium and nitrate analyzers)

**Answer 2:**
None

**Comment 3:**
Are substantial conclusions reached?
Based on the results conclusions can be reached; however the conclusions could be more specifically formulated. In the current version they are quite qualitative and not quantitative. For example: due to optimization with TOC, the original dosage can be reduced to approximately one third.

**Answer 3:**
It seems that Case 3 results, conclusion was not enough qualitative. It has been changed as follow to address that concern:
By having TOC analysis on-site and jar testing data with TOC and turbidity, plant operators did not have to wait for third party test results and could make immediate process decisions. The plant was able to save over $100k in chemicals and disposal costs and shown in Table 2. They also realized that effective TOC removal does not always correlate to effective turbidity removal or vice versa, therefore TOC and turbidity levels must both be monitored.

**Comment 4:**
Are the scientific methods and assumptions valid and clearly outlined?
Improvements can be made for this aspect. First of all, the method for each case is not structured in a distinct subsection (further elaborated at aspect no. 10.). The method for each section is clearly explained, but it is not compared with methods from other studies. Also explicit references to other works are not present in these paragraphs. (See also aspect no. 7.)

**Answer 4:**
For each of the 3 case studies, an improvement was made by creating a separate section for "Method and objective". In this section, authors clarified the overall subject of study, the previous monitoring (or non-monitoring situation), as well as why those operators were limited with their current methods. Then, the purpose of the study is described, as well as the objectives that were looked for. An effort was made to put the study objectives in perspective

with the operators challenges, so the content presents the usefulness of the monitoring in the industry.

**Comment 5:**
Are the results sufficient to support the interpretations and conclusions?
The reason for a selected period of the results shown should be made clear. Are these results a specific selection of a period or are these the results of a whole dataset? Also here, just like with the methods, a critical comparison with the results of previous or other studies is not made. (See also aspect no. 7.).

**Answer 5:**
For Case 1 and 2, a selection of results of a whole dataset was made. It appeared to the authors that presenting results in this format, was providing ability to visualize clearly fluctuations of TOC, that is to say the weekly and daily variations upon human activity (Case 1) or influence of coagulant change (for Case 2).

**Comment 6:**
Is the description of experiments and calculations sufficiently complete and precise to allow their reproduction by fellow scientists (traceability of results)?
The applied methods and measurement instruments are sufficiently described.

**Answer 6:**
None

**Comment 7:**
Do the authors give proper credit to related work and clearly indicate their own new/original contribution?
No. This is the major improvement to be made in this paper. No citations or references to related work are made in the text. Also a critical comparison of the results with other works is not made. It is advised to rewrite this paper in such a way that also the broader discussion on this topic becomes clear and also the position of this paper in this discussion.

**Answer 7:**
Great remark from referee, since the paper was missing the citations and references. After rewrite, the paper is now referencing previous work in the text and the references are listed at the end. We moved from 5 references quoted to 18 quoted and used inside the text to support the data and information provided. See also answer 5 for further details.

**Comment 8:**
Does the title clearly reflect the contents of the paper?
Yes. It clearly reflects the contents of the paper.

**Answer 8:**
None

**Comment 9:**
Does the abstract provide a concise and complete summary?
Yes. It covers the contents of the paper.

**Answer 9:**
None

**Comment 10:**
Is the overall presentation well structured and clear?

The main structure of the paper is clear. However, the structure for each case can be improved. In the current structure, there is a distiction between objective and results. However, in the subsection about results, also the methods are included.

For each case a clear distinction between methods and results should be made.

Section 2: The first two paragraphs of 2.2 are not results but methods. I would suggest to make an extra subsection 2.2 Methods, and subsection 2.3 with Results, and subsection 2.4 with Outlook (or futher investigation).

Section 3: In this section the distinction between objective, methods, results and outlook is not clear. Also here I would like to suggest to make a strict distinction between these subsections

Section 4: Idem dito.

**Answer 10:**
The structure of the paper was revamped to address that comment. Now, each of the 3 case has the following structure, with clear distinction:

1 Method and objective (see answer 4 for further details)

2 Results and further investigations

It was highly considered to explain the results from study, how do they apply for the plant operators and what further investigations could be made. Finally a quantification of results was added where missing (see also answer 3).

**Comment 11:**
Is the language fluent and precise?

Yes. The language is fluent and precise.

**Answer 11:**
None

**Comment 12:**
Are mathematical formulae, symbols, abbreviations, and units correctly defined and used?

The following abbreviations are not further defined: abstract: OPEX; section 2.2: SUMO.

**Answer 12:**
Sumo is actually not an abbreviation, but a name of a simulation program.

OPEX was explicated in text (Operational Expenditure)

**Comment 13:**
Should any parts of the paper (text, formulae, figures, tables) be clarified, reduced, combined, or eliminated?

The captions of the tables should be placed on top of the tables and not below the tables

**Answer 13:**
The captions of the tables were placed on top of the tables.

**Comment 14:**
Are the number and quality of references appropriate?

The number of references is too low and should be increased. Even more important, there are not any explicit citations or references within in text. There is only a References section included at the end.

**Answer 14:**
Great remark from referee, since the paper was missing the citations and references. After rewrite, the paper is now referencing previous work in the text and the references are listed at the end. We moved from 5 references quoted to 18 quoted and used inside the text to support the data and information provided. See also answer 5 for further details.

**Comment 15:**
Is the amount and quality of supplementary material appropriate?
Not available

**Answer 15:**
None